# Reciprocally Coupled Local Estimators Implement Bayesian Information Integration Distributively

**Wen-hao Zhang**[1,2,3], **Si Wu**[1]
[1]State Key Laboratory of Cognitive Neuroscience and Learning, and
IDG/McGovern Institute for Brain Research, Beijing Normal University, China.
[2]Institute of Neuroscience, Chinese Academy of Sciences, Shanghai, China.
[3]University of Chinese Academy of Sciences, Shanghai, China.
whzhang@ion.ac.cn,    wusi@bnu.edu.cn

## Abstract

Psychophysical experiments have demonstrated that the brain integrates information from multiple sensory cues in a near Bayesian optimal manner. The present study proposes a novel mechanism to achieve this. We consider two reciprocally connected networks, mimicking the integration of heading direction information between the dorsal medial superior temporal (MSTd) and the ventral intraparietal (VIP) areas. Each network serves as a local estimator and receives an independent cue, either the visual or the vestibular, as direct input for the external stimulus. We find that positive reciprocal interactions can improve the decoding accuracy of each individual network as if it implements Bayesian inference from two cues. Our model successfully explains the experimental finding that both MSTd and VIP achieve Bayesian multisensory integration, though each of them only receives a single cue as direct external input. Our result suggests that the brain may implement optimal information integration distributively at each local estimator through the reciprocal connections between cortical regions.

## 1   Introduction

In our daily life, we sense the world through multiple sensory systems. For instance, while walking, we perceive heading direction through either the visual cue (optic flow), or the vestibular cue generated by body movement, or both of them [1, 2]. In reality, because of noises, which arise due to signal ambiguity and/or fluctuations in neural transmission, our perception of the input information is often uncertain. In order to achieve an accurate or improved representation of the input information, it is critical for the brain to integrate information from multiple sensory modalities.

Mathematically, Bayesian inference provides an optimal way to estimate the stimulus value based on multiple uncertain information resources. Consider the task of inferring heading direction $\theta$ based on the visual and vestibular cues. Suppose that with a single cue $c_l$ ($l = \mathrm{vi}, \mathrm{ve}$ correspond to the visual and the vestibular cues, respectively), the estimation of heading direction satisfies the Gaussian distribution $p(c_l|\theta)$, which has the mean $\mu_l$ and the variance $\sigma_l^2$. Under the condition that noises from different cues are independent to each other, the Bayes' theorem states that

$$p(\theta|c_{\mathrm{vi}}, c_{\mathrm{ve}}) \propto p(c_{\mathrm{vi}}|\theta)p(c_{\mathrm{ve}}|\theta)p(\theta), \tag{1}$$

where $p(\theta|c_{\mathrm{vi}}, c_{\mathrm{ve}})$ is the posterior distribution of the stimulus when two cues are presented, and $p(\theta)$ the prior distribution. In the case of no prior knowledge, i.e., $p(\theta)$ is uniform, $p(\theta|c_{\mathrm{vi}}, c_{\mathrm{ve}})$ also

satisfies the Gaussian distribution with the mean and variance given by

$$\mu_b = \frac{\sigma_{\text{ve}}^2}{\sigma_{\text{vi}}^2 + \sigma_{\text{ve}}^2}\mu_{\text{vi}} + \frac{\sigma_{\text{vi}}^2}{\sigma_{\text{vi}}^2 + \sigma_{\text{ve}}^2}\mu_{\text{ve}}, \tag{2}$$

$$\frac{1}{\sigma_b^2} = \frac{1}{\sigma_{\text{vi}}^2} + \frac{1}{\sigma_{\text{ve}}^2}. \tag{3}$$

A number of elegant psychophysical experiments have demonstrated that humans and animals integrate multisensory information in an optimal Bayesian way. These include, for instances, using visual and auditory cues together to infer object location [3], getting the hand position from the visual and proprioceptive cues simultaneously [4], the combination of visual and haptic input to perceive object height [5], the integration of visual and vestibular cues to derive heading direction [6, 7], and the integration of texture and motion information to obtain depth [8]. Nevertheless, the detailed neural mechanism underlying Bayesian information integration remains largely unclear. Ma et. al., proposed a feed-forward mechanism to achieve Bayesian integration [9]. In their framework, a centralized network integrates information from multiple resources. In particular, in their model, the improved decoding accuracy after combining input cues (i.e., the decreased uncertainty given by Eq.3) depends on the linear response nature of neurons, a feature in accordance with the statistics of Poisson spiking train. However, it is unclear how well this result can be extended to non-Poisson statistics. Moreover, it is not clear where this centralized network responsible for information integration locates in the cortex.

In this work, we propose a novel mechanism to implement Bayesian information integration, which relies on the excitatory reciprocal interactions between local estimators, with each local estimator receiving an independent cue as external input. Although our idea may be applicable to general cases, the present study focuses on two reciprocally connected networks, mimicking the integration of heading direction information between the dorsal medial superior temporal (MSTd) area and ventral intraparietal (VIP) area. It is known that MSTd and VIP receive the visual and the vestibular cues as external input, respectively. We model each network as a continuous attractor neural network (CANN), reflecting the property that neurons in MSTd and VTP are widely tuned by heading direction [10, 11]. Interestingly, we find that with positive reciprocal interactions, both networks read out heading direction optimally in Bayesian sense, despite the fact that each network only receives a single cue as directly external input. This agrees well with the experimental finding that both MSTd and VIP integrate the visual and vestibular cues optimally [6, 7]. Our result suggests that the brain may implement Bayesian information integration distributively at each local area through reciprocal connections between cortical regions.

## 2 The Model

We consider two reciprocally connected networks, each of which receives the stimulus information from an independent sensory cue (see Fig.1A). The two networks may be regarded as representing, respectively, the neural circuits in MSTd and VIP. Anatomical and fMRI data have revealed that there exist abundant reciprocal interactions between MSTd and VIP [12–14]. Neurons in MSTd and VIP are tuned to heading direction, relying on the visual and the vestibular cues [10, 15].

CANNs, also known as neural field model, have been successfully applied to describe the encoding of head-direction in neural systems [16]. Therefore, we build each network as a CANN. Denote $\theta$ to be the stimulus value (i.e. the heading direction) encoded by both networks, and the neuronal preferred stimuli are in the range of $-\pi < \theta \leq \pi$ with periodic boundary condition. Denote $U_l(\theta, t)$, for $l = 1, 2$, the synaptic input at time $t$ to the neurons having the preferred stimulus $\theta$ in the $l$-th network. The dynamics of $U_l(\theta, t)$ is determined by the recurrent inputs from other neurons in the same network, the reciprocal inputs from neurons in the other network, the external input $I_l^{ext}(\theta, t)$, and its own relaxation. It is written as

$$\tau \frac{\partial}{\partial t} \begin{bmatrix} U_1(\theta,t) \\ U_2(\theta,t) \end{bmatrix} = \rho \int \begin{bmatrix} W_{11} & W_{12} \\ W_{21} & W_{22} \end{bmatrix} \begin{bmatrix} r_1(\theta',t) \\ r_2(\theta',t) \end{bmatrix} d\theta' + \begin{bmatrix} I_1^{ext}(\theta,t) \\ I_2^{ext}(\theta,t) \end{bmatrix} - \begin{bmatrix} U_1(\theta,t) \\ U_2(\theta,t) \end{bmatrix}, \tag{4}$$

where $\tau$ is the time constant for synaptic current, which is typically in the order of 2-5ms. $\rho$ is the neural density. $r_l(\theta, t)$ is the firing rate of neurons, which increases with the synaptic input but saturates when the synaptic input is sufficiently large. The saturation is mediated by the contribution

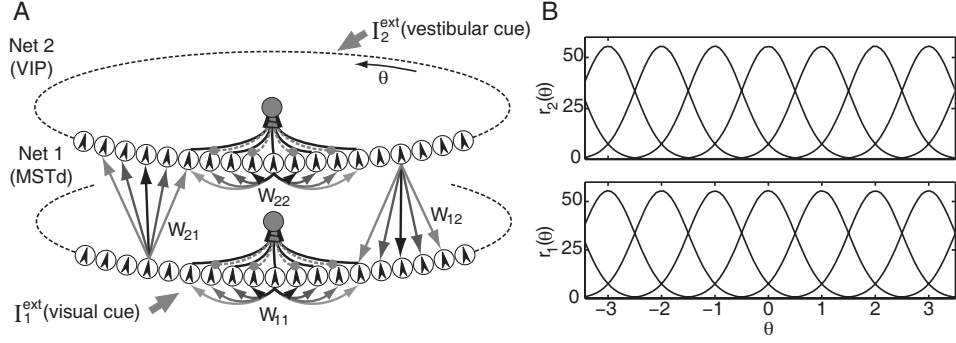

Figure 1: Network structure and stationary state. (A). The two networks are reciprocally connected and each of them forms a CANN. Each disk represents an excitatory neuron with its preferred heading direction indicated by the arrow inside. The gray disk in the middle of the network represents the inhibitory neuron pool which sums the total activities of excitatory neurons and generates divisive normalization (Eq.5). The solid line with arrow is excitatory connection with the gray level indicating the strength. The gray dashed line with dots represents inhibitory connection. (B). The stationary states of two networks, which can locate at anywhere in the perceptual space. Parameters: $N = 100$, $k = 10^{-3}$, $a = 0.5$, $L = 7$, $J_{11} = J_{22} = 1.5J_c$, $J_{12} = J_{21} = 0.5J_{11}$.

of inhibitory neurons not explicitly presented in our framework. A solvable model captures these features is given by divisive normalization [17, 18],

$$r_l(\theta, t) = \frac{[U_l(\theta, t)]_+^2}{1 + k\rho \int_{\theta'} [U_l(\theta', t)]_+^2 \, d\theta'}, \tag{5}$$

where the symbol $[x]_+$ denotes a half-rectifying function, i.e., $[x]_+ = 0$, for $x \leq 0$ and $[x]_+ = x$, for $x > 0$, and $k$ reflects the strength of global inhibition.

$W_{lm}(\theta, \theta')$ denotes the connection from the neurons $\theta'$ in the network $m$ to the neurons $\theta$ in the network $l$. $W_{11}(\theta, \theta')$ and $W_{22}(\theta, \theta')$ are the recurrent connections within the same network, and $W_{12}(\theta, \theta')$ and $W_{21}(\theta, \theta')$ the reciprocal connections between the networks. We assume they are of the Gaussian form, i.e.,

$$W_{lm}(\theta, \theta') = \frac{J_{lm}}{\sqrt{2\pi}a_{lm}} \exp\left[-\frac{(\theta - \theta')^2}{2a_{lm}^2}\right], \tag{6}$$

where $a_{lm}$ determines the neuronal interaction range. In the text below, we consider $a_{lm} \ll \pi$ and effectively take $-\infty < \theta < \infty$ in the theoretical analysis. We choose $J_{lm} > 0$, for $l, m = 1, 2$, implying excitatory recurrent and reciprocal neuronal interactions. The contribution of inhibitory neurons is implicitly included in the divisive normalization.

The external inputs to two networks are given by

$$I_l^{ext}(\theta, t) = \alpha_l \exp\left[-\frac{(\theta - \mu_l)^2}{4(a_{ll})^2}\right] + \eta_l \xi_l(\theta, t), \tag{7}$$

where $\mu_l$ denotes the stimulus value conveyed to the network $l$ by the corresponding sensory cue. This can be understood as $I_l^{ext}$ drives the network $l$ to be stable at $\mu_l$ when no reciprocal interaction and noise exist. $\alpha_l$ is the input strength, and $\xi_l(\theta, t)$ is the Gaussian white noise of zero mean and unit variance, with $\eta_l$ the noise amplitude. The noise term causes the uncertainty of the input information, which induces fluctuations of the network state. The exact form of $I_l^{ext}$ is not critical here, as long as it has an unimodal form.

## 2.1 The dynamics of uncoupled networks

It is instructive to first review the dynamics of two networks without reciprocal connection (by setting $W_{lm} = 0$ for $l \neq m$ in Eq.4). In this case, the dynamics of each network is independent

from the other. Because of the translation-invariance of the recurrent connections $W_{ll}(\theta, \theta')$, each network can support a continuous family of active stationary states even when the external input is removed [19]. These attractor states are of Gaussian-shape, called bumps, which are given by,

$$\tilde{U}_l(x) \quad = \quad U_l^0 \exp\left[-\frac{(\theta - z_l)^2}{4(a_{ll})^2}\right], \tag{8}$$

where $z_l$ is a free parameter, representing the peak position of the bump, and $U_l^0 = [1 + (1 - J_c/J_{ll})^{1/2}]J_{ll}/(4a_{ll}k\sqrt{\pi})$. The bumps are stable for $J_{ll} > J_c$, with $J_c = 2\sqrt{2}(2\pi)^{1/4}\sqrt{ka_{ll}/\rho}$, the critical connection strength below which only silent states, $U_l^0 = 0$, exist.

In response to external inputs, the bump position $z_l$ is interpreted as the population decoding result of the network. It has been proven that for a strong transient or a weak constant input, the network bump will move toward and be stable at a position having the maximum overlap with the noisy input, realizing the so called template-matching operation [17, 18]. For temporally fluctuating inputs, the bump position also fluctuates in time, and the variance of bump position measures the network decoding uncertainty.

In a CANN, its stationary states form a continuous manifold in which the network is neutrally stable, i.e., the network state can translate smoothly when the external input changes continuously [18, 20]. This neutral stability is the key that enables the neural system to track moving direction, head-direction and spatial location of objects smoothly [16, 21, 22]. Due to the special structure of a CANN, it has been proved that the dynamics of a CANN is dominated by a few motion modes, corresponding to distortions in the height, position and other higher order features of the Gaussian bump [19]. In the weak input limit, it is enough to project the network dynamics onto the first few dominating motion modes and neglect the higher order ones then simplify the network dynamics significantly. The first two dominating motion modes we are going to use are,

$$\text{height}: \phi_0(\theta|z) \quad = \quad \exp\left[-\frac{(\theta - z)^2}{4a^2}\right], \tag{9}$$

$$\text{position}: \phi_1(\theta|z) \quad = \quad \left(\frac{\theta - z}{a}\right)\exp\left[-\frac{(\theta - z)^2}{4a^2}\right], \tag{10}$$

where $a$ is the width of the basis function, whose value is determined by the bump width the network holds. By projecting a function $f(\theta)$ on a motion mode $\phi(\theta|z)$, we mean to compute the quantity, $\int_\theta f(\theta)\phi(\theta|z)d\theta / \int_\theta \phi(\theta|z)d\theta$.

When reciprocal connections are included, the dynamics of the two networks interact with each other. The bump position of each network is no longer solely determined by its own input, but is also affected by the input to the other network, enabling both networks to integrate two sensory cues via reciprocal connections. We consider the reciprocal connections, $W_{lm}(\theta, \theta')$, for $l \neq m$, also translation-invariant (Eq.6), so that two networks still hold the key property of CANNs. That is, they can hold a continuous family of stationary states and track time-varying inputs smoothly (Fig.1B).

## 3   Dynamics of Coupled Networks

It is in general difficult to analyze the dynamics of two coupled networks. In the text below, we will consider the weak input limit and use a projection method to simplify the network dynamics. The simplified model allows us to solve the network decoding performances analytically and gives us insight into the understanding of how reciprocal connections help both networks to integrate information optimally from independent cues.

For simplicity, we consider two networks that are completely symmetric, i.e., they have the same structure, i.e., $J_{11} = J_{22} \equiv J_{\text{rc}}$, $J_{12} = J_{21} \equiv J_{\text{rp}}$, and $a_{lm} = a$; and they receive the same mean input value and input strength, i.e., $\mu_1 = \mu_2 \equiv \mu$, $\alpha_1 = \alpha_2 \equiv \alpha$ and $\eta_1 = \eta_2 \equiv \eta$. They receive, however, independent noises, i.e., $\langle \xi_1 \xi_2 \rangle = 0$, implying that two cues are independent to each other given the stimulus.

In the weak input limit (i.e., for small enough $\alpha$), we find that the network states have approximately Gaussian shape and their variations are dominated by the height and position changes of the bump

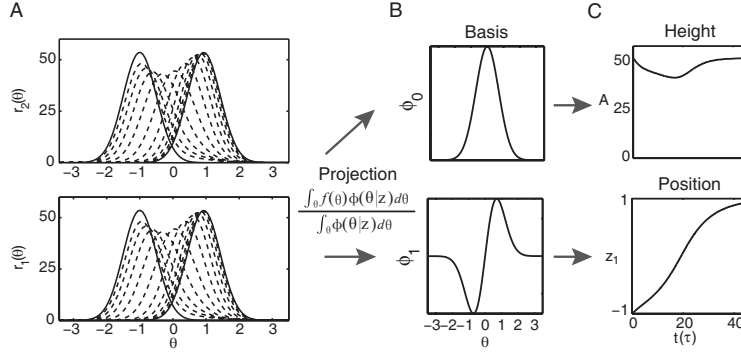

Figure 2: Characters of the network dynamics. (A). Two networks receive the same external input, whose value jumps from $-1$ to $1$ abruptly. The network states move smoothly from the initial to the target position, and their main changes are the height and the position of the Gaussian bumps. (B) The basis functions for the two dominating motion modes. (C) The simplified network dynamics after projecting onto the two dominating motion modes. Parameters: $\alpha_1 = \alpha_2 = 0.2U^0$, $\eta_1 = \eta_2 = 0$, and others are the same as Fig.1.

(see Fig.2). Thus, we take the Gaussian ansatz and assume the network states to be

$$U_l(\theta, t) \approx A(t)\exp\left[-\frac{(\theta - z_l(t))^2}{4a^2}\right], \tag{11}$$

$$r_l(\theta, t) \approx B(t)\exp\left[-\frac{(\theta - z_l(t))^2}{2a^2}\right], \tag{12}$$

where $A(t)$ represents the bump height, $z(t)$ the bump position in the network $l$, $a$ the bump width and $B = [A]_+^2/(1 + \sqrt{2\pi}k\rho a[A]_+^2)$ according to Eq.(5). Note that the bumps in two networks have the same shape but different positions due to independent noises.

Substituting Eqs.(11,12) and (7) into the network dynamics Eq.(4), and projecting them onto the height and position motion modes (9-10), we obtain the dynamics for the height and position of the bumps in two networks (see Supplemental information 1), which are

$$\tau\frac{dA}{dt} = -A + (\tilde{J}_{\rm rc} + \tilde{J}_{\rm rp})B + \alpha, \tag{13}$$

$$\tau\frac{dz_1}{dt} = \frac{\tilde{J}_{\rm rp}B}{A}(z_2 - z_1) + \frac{\alpha}{A}(\mu - z_1) + \frac{2\eta\sqrt{a}}{(2\pi)^{1/4}A}\xi_1(t), \tag{14}$$

$$\tau\frac{dz_2}{dt} = \frac{\tilde{J}_{\rm rp}B}{A}(z_1 - z_2) + \frac{\alpha}{A}(\mu - z_2) + \frac{2\eta\sqrt{a}}{(2\pi)^{1/4}A}\xi_2(t), \tag{15}$$

where $\tilde{J} \equiv \rho J/\sqrt{2}$ for simplifying notation. By removing the external inputs (by setting $\alpha = 0$ in Eq.(13)), we can get the necessary condition for the networks holding self-sustained bump states, which is (see Supplemental information 2)

$$J_{\rm rc} + J_{\rm rp} \geq 2\sqrt{2}(2\pi)^{1/4}\sqrt{ka/\rho}. \tag{16}$$

It indicates that positive reciprocal interactions $J_{rp}$ help the networks to retain attractor states.

To get clear understanding of the effect of reciprocal connections, we decouple the dynamics of $z_1$ and $z_2$ by studying the dynamics of their their difference, $z_d = z_1 - z_2$, and their summation, $z_s = z_1 + z_2$. From Eqs.(14) and (15), we obtain

$$\tau\frac{dz_d}{dt} = -\frac{\alpha + 2\tilde{J}_{\rm rp}B}{A}z_d + \frac{2\sqrt{2}\eta\sqrt{a}}{(2\pi)^{1/4}A}\epsilon_d(t), \tag{17}$$

$$\tau\frac{dz_s}{dt} = -\frac{\alpha}{A}z_s + \frac{2\alpha}{A}\mu + \frac{2\sqrt{2}\eta\sqrt{a}}{(2\pi)^{1/4}A}\epsilon_s(t), \tag{18}$$

where $\epsilon_d(t)$ and $\epsilon_s(t)$ are independent Gaussian white noises re-organized from $\xi_1(t)$ and $\xi_2(t)$ ($\sqrt{2}\epsilon = \xi_1 \pm \xi_2$).

By solving the above stochastic differential equations, we get the means and variances of $z_d$ and $z_s$ in the limit of $t = \infty$, which are

$$\langle z_d \rangle = 0, \quad \langle z_s \rangle = 2\mu, \tag{19}$$

$$\mathrm{Var}(z_d) \equiv \langle (z_d - \langle z_d \rangle)^2 \rangle = \frac{4\eta^2 a}{\sqrt{2\pi}\tau A} \frac{1}{\alpha + 2\tilde{J}_{\mathrm{rp}}B}, \tag{20}$$

$$\mathrm{Var}(z_s) \equiv \langle (z_s - \langle z_s \rangle)^2 \rangle = \frac{4\eta^2 a}{\sqrt{2\pi}\tau A\alpha}, \tag{21}$$

where the symbol $\langle \cdot \rangle$ represents averaging over many trails. Eq.(20) indicates that the positive reciprocal connections $\tilde{J}_{rp}$ tend to decrease the variance of $z_d$, i.e, the difference between the states in two networks (in practice, varying $\tilde{J}_{rp}$ also induces mild changes in $A$, $B$ and $a$; we have confirmed in simulation that for a wide range of parameters, increasing $\tilde{J}_{rp}$ indeed decrease $\mathrm{Var}(z_d)$).

The decoding error of each network, measured by the variance of $z_l$, is calculated to be (two networks have the same result due to the symmetry),

$$\langle z_l \rangle = \mu, \quad \text{for} \quad l = 1, 2, \tag{22}$$

$$\mathrm{Var}(z_l) = [\mathrm{Var}(z_d) + \mathrm{Var}(z_s)]/4,$$

$$= \frac{\eta^2 a}{\sqrt{2\pi}\tau A} \left( \frac{1}{\alpha} + \frac{1}{\alpha + 2\tilde{J}_{\mathrm{rp}}B} \right). \tag{23}$$

We see that the network decoding is unbiased and their errors tend to decrease with the reciprocal connection strength $\tilde{J}_{rp}$ (see the second term in the right-hand of Eq.23). It is easy to check that in the extreme cases and assuming the bump shape is unchanged (which is not true but is still a good indication), the network decoding variance with vanishing reciprocal interaction ($\tilde{J}_{rp} = 0$) is twofold of that with infinitely strong reciprocal interactions ($\tilde{J}_{rp} = \infty$). Thus, reciprocal connections between networks do provide an effective way to integrate information from independent input cues.

To further display the advantage of reciprocal connection, we also calculate the situation when a single network receives both input cues. This equals to setting the external input to a single CANN to be $I^{ext}(x, t) = 2\alpha e^{-(x-\mu)^2/4a^2} + \sqrt{2}\eta\xi(x, t)$ (see Eq.(7) and consider the independence between two cues). The result in this case can be obtained straightforwardly from Eq.(23) by choosing $\tilde{J}_{\mathrm{rp}} = 0$ and replacing $\eta$ with $\sqrt{2}\eta$ and $\alpha$ with $2\alpha$, which gives $\mathrm{Var}(z)_{\mathrm{single}} = 2\eta^2 a/(\sqrt{2\pi}\tau A\alpha)$. This result equals to the error when two networks are uncoupled and is larger than that of the coupled case.

In the weak input limit, the decoding errors in general situations when two networks are not symmetric can also be calculated, (see Supplemental information 3)

$$\mathrm{Var}(z_1) = \frac{2a}{\sqrt{2\pi}\tau} \frac{[(\tilde{J}_{12}B_2\alpha_2 + \tilde{J}_{21}B_1\alpha_1 + \alpha_1\alpha_2)A_2/A_1 + (\tilde{J}_{21}B_1 + \alpha_2)^2]\eta_1^2 + (\tilde{J}_{12}B_2)^2\eta_2^2}{(\tilde{J}_{12}B_2A_2 + \alpha_1A_2 + \tilde{J}_{21}B_1A_1 + \alpha_2A_1)(\tilde{J}_{12}B_2\alpha_2 + \tilde{J}_{21}B_1\alpha_1 + \alpha_1\alpha_2)}. \tag{24}$$

$\mathrm{Var}(z_2)$ has the same form as $\mathrm{Var}(z_1)$ except that the indexes 1 and 2 are interchanged.

## 4 Coupled Networks Implement Bayesian Information Integration

In this section, we compare the network performances with experimental findings. Mimicking the experimental setting for exploring the integration of visual and vestibular cues in the inference of heading direction [6, 7], we apply three input conditions to two networks (see Fig.3A), which are:

- Only visual cue: $\quad \alpha_1 = \alpha, \quad \alpha_2 = 0.$
- Only vestibular cue: $\quad \alpha_1 = 0, \quad \alpha_2 = \alpha.$
- Combined cues: $\quad \alpha_1 = \alpha, \quad \alpha_2 = \alpha.$

In three conditions, the noise amplitude is unchanged and the reciprocal connections are intact.

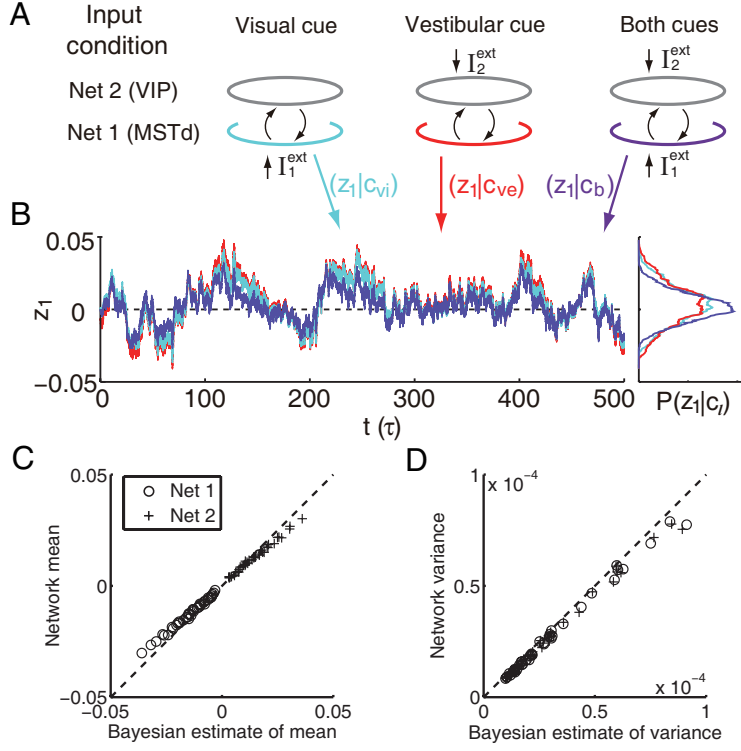

Figure 3: Two coupled-networks implement (nearly) Bayesian inference. (A). The three input conditions to two networks. (B). The bump position of the network 1 fluctuates around the true stimulus value 0. The right panel displays the bump position distributions in three input conditions, from which we estimate the mean and variance of the decoding results. (C),(D). Compared the network decoding results with two cues with the predictions of Bayesian inference. (C) for the mean value and (D) for the variance. Different combinations of the input strengths $\alpha_l$ and the reciprocal connection strengths $J_{rp}$ are chosen. Parameters: $\mu_1 = -0.07, \mu_2 = 0.07, \eta_1 = \eta_2 = 0.5$, $\alpha_i \in [0.1, 0.5]U^0$, $J_{rp} \in [0.3, 1]J_{rc}$, and the others are the same as Fig.1.

Considering the symmetric structures of two networks and ignoring the mild changes in the bump shape in the weak input limit, we can obtain from Eq.(24) the decoding variance in the three input conditions, which are (because of the symmetry, only the results for the network 1 are shown)

$$\text{Var}(z_1|c_{vi}) = \frac{2a\eta^2}{\sqrt{2\pi}\tau\alpha A}, \tag{25}$$

$$\text{Var}(z_1|c_{ve}) = \frac{2a\eta^2}{\sqrt{2\pi}\tau\alpha A}\frac{\tilde{J}_{rp}B + \alpha}{\tilde{J}_{rp}B}, \tag{26}$$

$$\text{Var}(z_1|c_b) = \frac{2a\eta^2}{\sqrt{2\pi}\tau\alpha A}\frac{\tilde{J}_{rp}B + \alpha}{2\tilde{J}_{rp}B + \alpha}, \tag{27}$$

where $\text{Var}(z_1|c_{vi})$, $\text{Var}(z_1|c_{ve})$ and $\text{Var}(z_1|c_b)$ denote, respectively, the decoding errors when only the visual cue, only the vestibular cue and both cues are presented. It is straightforward to check that

$$\frac{1}{\text{Var}(z_1|c_b)} = \frac{1}{\text{Var}(z_1|c_{vi})} + \frac{1}{\text{Var}(z_1|c_{ve})}. \tag{28}$$

Thus, in the weak input limit, the coupled CANNs implements Bayesian inference perfectly (compare Eq.(28) to the Bayesian criterion Eq.(3)).

We carry out simulations to further confirm the above theoretical analysis. We run the network dynamics under three input conditions for many trials, and calculate the means and variances of the bump positions in each condition. Fig.3B shows that the bump position fluctuations become

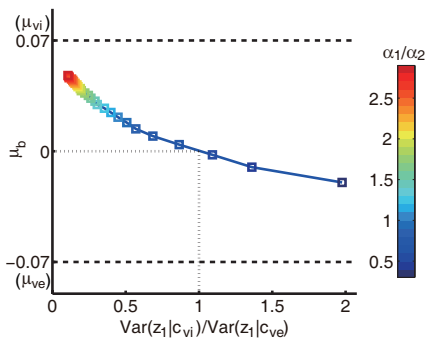

Figure 4: The decoding mean of the network 1 shifts toward the more reliable cue. The color encodes the ratio of the input strengths to two networks $\alpha_1/\alpha_2$, which generates varied reliability for two cues. For increasing the ratio $\mathrm{Var}(z_1|c_{\mathrm{vi}})/\mathrm{Var}(z_1|c_{\mathrm{ve}})$, i.e., the vestibular cue becomes more reliable than the visual one, the network estimation shifts to the stimulus value $\mu_2$ conveyed by the vestibular cue. Parameters: $\mu_1 = 0.07$, $\mu_2 = -0.07$, $\eta_1 = \eta_2 = 0.01$ and the others are the same as Fig.1.

narrower in the combined cue input condition, indicating greater accuracy in the decoding. We compare the result when both cues are presented with the prediction of the Bayesian inference, obtained by using Eqs.(2, 3). Fig.3C and D show that two networks indeed achieve near Bayesian optimal inference for a wide range of input amplitudes and reciprocal connection strengths.

A salient feature of Bayesian inference is that its decoding value is biased to the more reliable cue. The reliability of cues is quantified by their variance ratio, e.g., $(\sigma_{\mathrm{vi}})^2 < (\sigma_{\mathrm{ve}})^2$ means that visual cue is more reliable than vestibular cue. From Eq.2, we see that Bayesian inference gives a larger weight to the more reliable cue. This property has been used as a criterion in experiment to check the implementation of Bayesian inference, called "reliability based cue weighting" [23]. We also test this property in our model. To achieve different reliability of the cues, we adjust the input strength $\alpha_1$, and keep the other input parameters unchanged, mimicking the experimental finding that the firing rate of MT neuron, the earlier stage before MSTd, increases with the input coherence for its preferred stimuli [24]. With varying input strengths $\alpha_1$, and hence varied ratios $\mathrm{Var}(z_1|c_{\mathrm{vi}})/\mathrm{Var}(z_1|c_{\mathrm{ve}})$, we calculate the mean of the network decoding. Fig. 4 shows that the decoded mean in the combined cues condition indeed shifts towards to the more reliable cue, agreeing with the experimental finding and the property of Bayesian inference.

## 5 Conclusion

In the present study, we have proposed a novel mechanism to implement Bayesian information integration. We consider two networks which are reciprocally connected, and each of them is modeled as a CANN receiving the stimulus information from an independent cue. Our network model may be regarded as mimicking the information integration on heading direction between the neural circuits in MSTd and VIP. Experimental data has revealed that the two areas are densely connected in reciprocity and that neurons in both areas are widely tuned by heading direction, favoring our model assumptions.

We use a projection method to solve the network dynamics in the weak input limit analytically and get insights into how positive reciprocal connections enable one network to effectively integrate information from the other. We then carry out simulations to confirm the theoretical analysis, following the experimental protocols. Our results show that both networks realize near Bayesian optimal decoding for a wide range of parameters, supporting the experimental finding that both MSTd and VIP optimally integrate the visual and the vestibular cues in heading direction inference, though each of them only receives a single cue directly.

Our study may have a far-reaching implication on neural information processing. It suggests that the brain can implement efficient information integration in a distributive manner through reciprocal connections between cortical regions. Compared to centralized information integration, distributive processing is more robust to local failures and facilitates parallel computation.

## 6 Acknowledgements

This work is supported by National Foundation of Natural Science of China (No.91132702 and No.31261160495).

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
