[Supplementary Material]

# Reciprocally Coupled Local Estimators Implement Bayesian Information Integration: Supplemental Information

**Wen-hao Zhang**[1,2,3], **Si Wu**[1]
[1]State Key Laboratory of Cognitive Neuroscience and Learning, and
IDG/McGovern Institute for Brain Research, Beijing Normal University, China.
[2]Institute of Neuroscience, Chinese Academy of Sciences, Shanghai, China.
[3]University of Chinese Academy of Sciences, Shanghai, China.
whzhang@ion.ac.cn,   wusi@bnu.edu.cn

## 1 The Simplified Network Dynamics

Substituting the Gaussian ansatz Eqs.(11,12) into the network dynamics Eq.(4), and under the condition of equal connection width, we obtain the left-hand (LHS) and the right-hand sides (RHS) of Eq.(4) (only the result for the network 1 is shown; the result for the network 2 can be similar obtained), which are:

$$
\text{LHS} = \tau \frac{dA_1}{dt} \exp\left[-\frac{(\theta - z_1)^2}{4a^2}\right] + \frac{\tau A_1}{2a} \frac{dz_1}{dt} \left(\frac{\theta - z_1}{a}\right) \exp\left[-\frac{(\theta - z_1)^2}{4a^2}\right]. \tag{S1}
$$

$$
\begin{aligned}
\text{RHS} &= -A_1 \exp\left[-\frac{(\theta - z_1)^2}{4a^2}\right] + \rho \int_{\theta'} \frac{J_{11}B_1}{\sqrt{2\pi}a} \exp\left[-\frac{(\theta - \theta')^2}{2a^2}\right] \exp\left[-\frac{(\theta' - z_1)^2}{2a^2}\right] d\theta' \\
&\quad + \rho \int_{\theta'} \frac{J_{12}B_2}{\sqrt{2\pi}a} \exp\left[-\frac{(\theta - \theta')^2}{2a^2}\right] \exp\left[-\frac{(\theta' - z_2)^2}{2a^2}\right] d\theta' + \alpha_1 \exp\left[-\frac{(\theta - \mu_1)^2}{4a^2}\right] + \eta_1 \xi_1(\theta, t), \\
&= -A_1 \exp\left[-\frac{(\theta - z_1)^2}{4a^2}\right] + \frac{\rho J_{11}B_1}{\sqrt{2}} \exp\left[-\frac{(\theta - z_1)^2}{2a^2}\right], \\
&\quad + \frac{\rho J_{12}B_2}{\sqrt{2}} \exp\left[-\frac{(\theta - z_2)^2}{2a^2}\right] + \alpha_1 \exp\left[-\frac{(\theta - \mu_1)^2}{4a^2}\right] + \eta_1 \xi_1(\theta, t). 
\end{aligned} \tag{S2}
$$

Projecting Eqs.(S1-S2) onto the height and position motion modes (Eq. 9 and 10), we get the simplified dynamics,

$$
\begin{aligned}
\tau \frac{dA_1}{dt} &= -A_1 + \frac{\rho J_{11}B_1}{\sqrt{2}} + \frac{\rho J_{12}B_2}{\sqrt{2}} \exp\left[-\frac{(z_1 - z_2)^2}{8a^2}\right], \\
&\quad + \alpha_1 \exp\left[-\frac{(\mu_1 - z_1)^2}{8a^2}\right]
\end{aligned} \tag{S3}
$$

$$
\begin{aligned}
\frac{\tau A_1}{2a} \frac{dz_1}{dt} &= \frac{\rho J_{12}B_2}{2\sqrt{2}a} (z_2 - z_1) \exp\left[-\frac{(z_1 - z_2)^2}{8a^2}\right] \\
&\quad + \frac{\alpha_1}{2a} (\mu_1 - z_1) \exp\left[-\frac{(\mu_1 - z_1)^2}{8a^2}\right] + \frac{2\eta_1 \sqrt{a}}{(2\pi)^{1/4}A_1} \xi_1(t). 
\end{aligned} \tag{S4}
$$

When $(z_1 - z_2)^2/8a^2$ is sufficiently small (which is the case for the parameters we choose), the above equations can be further simplified (the dynamics for the network 2 is added),

$$\tau \frac{dA_1}{dt} = -A_1 + \frac{\rho J_{11} B_1}{\sqrt{2}} + \frac{\rho J_{12} B_2}{\sqrt{2}} + \alpha_1, \tag{S5}$$

$$\tau \frac{dz_1}{dt} = \frac{\rho J_{12} B_2}{\sqrt{2} A_1} (z_2 - z_1) + \frac{\alpha_1}{A_1}(\mu_1 - z_1) + \frac{2\eta_1 \sqrt{a}}{(2\pi)^{1/4} A_1} \xi_1(t), \tag{S6}$$

$$\tau \frac{dA_2}{dt} = -A_2 + \frac{\rho J_{22} B_2}{\sqrt{2}} + \frac{\rho J_{21} B_1}{\sqrt{2}} + \alpha_2, \tag{S7}$$

$$\tau \frac{dz_2}{dt} = \frac{\rho J_{21} B_1}{\sqrt{2} A_2} (z_1 - z_2) + \frac{\alpha_2}{A_2}(\mu_2 - z_2) + \frac{2\eta_2 \sqrt{a}}{(2\pi)^{1/4} A_2} \xi_2(t). \tag{S8}$$

When two networks are completely symmetric and replace $\rho J_{lm}/\sqrt{2}$ with $\tilde{J}_{lm}$, the above equations give to Eqs.(13,14,15) in the main text.

For the convenience of description, we re-organize the position dynamics into the matrix form, which is given by,

$$\dot{\mathbf{Z}} = \mathbf{M}_Z \mathbf{Z} + \mathbf{I}_Z + \boldsymbol{\beta}\xi(t), \tag{S9}$$

with $\mathbf{Z} = (z_1, z_2)^{\mathrm{T}}$, $\mathbf{I}_Z = (\alpha_1\mu_1/\tau A_1, \alpha_2\mu_2/\tau A_2)^{\mathrm{T}}$, $\boldsymbol{\beta} = (\beta_1, \beta_2)^{\mathrm{T}} = \left(\frac{2\eta_1 \sqrt{a}}{(2\pi)^{1/4}\tau A_1}, \frac{2\eta_2 \sqrt{a}}{(2\pi)^{1/4}\tau A_2}\right)^{\mathrm{T}}$, and

$$\mathbf{M}_Z = \frac{1}{\tau} \begin{bmatrix} -\frac{\tilde{J}_{12} B_2 + \alpha_1}{A_1} & \frac{\tilde{J}_{12} B_2}{A_1} \\ \frac{\tilde{J}_{21} B_1}{A_2} & -\frac{\tilde{J}_{21} B_1 + \alpha_2}{A_2} \end{bmatrix}. \tag{S10}$$

## 2  Stationary States of the Coupled Networks

When two networks are completely symmetric, without external inputs, the bump heights for two networks are exactly the same, which satisfy (from Eq.13)

$$A = (J_{rc} + J_{rp})\rho B/\sqrt{2},$$
$$= (J_{rc} + J_{rp})\frac{\rho A_2}{\sqrt{2}(1 + \sqrt{2\pi}k\rho a A_2)}.$$

After some algebra, we get

$$A\left[\sqrt{2\pi}k\rho a A_2 - \frac{\rho(J_{rc} + J_{rp})}{\sqrt{2}} A + 1\right] = 0. \tag{S11}$$

Thus, the steady active bump height (non-zero) is solved to be

$$A = \frac{(J_{rc} + J_{rp})}{4\sqrt{\pi}ka}\left[1 + \sqrt{1 - \frac{8\sqrt{2\pi}ka}{\rho(J_{rc} + J_{rp})^2}}\right]. \tag{S12}$$

The necessary condition for existence of active bump state is therefore given by

$$J_{rc} + J_{rp} \geq 2\sqrt{2}(2\pi)^{1/4}\sqrt{ka/\rho}. \tag{S13}$$

## 3  Decoding Results of the Coupled Networks

If the parameters for two networks are not the same, the decoding errors can be obtained by solving the stochastic differential equation (Eq.S9) directly, which is given by

$$\mathbf{Z}(t) = e^{\mathbf{M}_Z t}\left[\mathbf{Z}(0) + \int_0^t e^{-\mathbf{M}_Z s}\mathbf{I}_Z(s)ds + \int_0^t e^{-\mathbf{M}_Z s}\boldsymbol{\beta}d\mathbf{W_s}\right], \tag{S14}$$

where $\mathbf{W_s}$ is a 2-by-1 vector and each element denoting a standard Wiener process, and $\langle d\mathbf{W_s}(i) d\mathbf{W_s}(j) \rangle = \delta(i-j)$. From the solution of $\mathbf{Z}(t)$, the dynamics of covariance matrix of bump positions $\mathrm{Cov}[\mathbf{Z}(t)]$ is governed by,

$$\frac{d\mathrm{Cov}[\mathbf{Z}(t)]}{dt} = \mathbf{M}_Z \mathrm{Cov}[\mathbf{Z}(t)] + (\mathbf{M}_Z \mathrm{Cov}[\mathbf{Z}(t)])^{\mathrm{T}} + \boldsymbol{\beta}\boldsymbol{\beta}^{\mathrm{T}}. \tag{S15}$$

The stationary value of $\mathrm{Cov}[Z(t)]$ satisfies

$$\mathbf{M}_Z \mathrm{Cov}[\mathbf{Z}(t)] + (\mathbf{M}_Z \mathrm{Cov}[\mathbf{Z}(t)])^{\mathrm{T}} = \boldsymbol{\beta}\boldsymbol{\beta}^{\mathrm{T}}. \tag{S16}$$

Pairing elements by elements of the above equation, we get

$$\begin{aligned} m_{11}V_{11} + m_{12}V_{21} &= \beta_1^2/2, \\ m_{22}V_{22} + m_{21}V_{12} &= \beta_2^2/2, \\ m_{12}V_{22} + m_{21}V_{11} &= -T_Z V_{12}. \end{aligned}$$

After some algebra, we get the decoding variances

$$\begin{pmatrix} V_{11} \\ V_{22} \end{pmatrix} = -\frac{1}{2T_Z D_Z} \begin{pmatrix} (D_Z + m_{22}^2)\beta_1^2 + m_{12}^2\beta_2^2 \\ m_{21}^2\beta_1^2 + (D_Z + m_{11}^2)\beta_2^2 \end{pmatrix}, \tag{S17}$$

where $T_z$ and $D_Z$ are the trace and determinant of the matrix $\mathbf{M}_Z$, respectively. Substituting the elements of $\mathbf{M}_Z$ into above equation, we obtain the Eq.24.