[Reviews · NeurIPS 2013]

Submitted by Assigned_Reviewer_1

The authors show by approximate analysis of two identical continuous attractor networks (Zhang 1996), reciprocally coupled by Gaussian weights, that such a network can approximately implement the Bayesian posterior solution for queue integration.

General comment:

The analysis and the basic finding for the weak input regime seems solid and original. However, the model requires a lot of machinery for computing equations (2,3) and also the connection to neuroscience is rather week.

Specific comments/questions:

How realistic is the weak input regime for biology? For example, how long does it take after input onset to establish a bump reflecting the correct stimulus value?

In Fig 3 B (right panel): Why appear the bump positions from visual or vestibular cue alone to coincide? The caption says that \mu_2 set to be offset of each other and also the bump reflecting the combined inputs is offset.

The comparison to biological data is only qualitative here. A quantitative comparison would make the paper stronger.

The authors should discuss the possibility of a neural implementation of their model, for example, the model brittleness due to the fine tuning requirements of CANNs.

Discuss the reciprocal wiring requirements: There seem to be more reciprocal wires required than there are neurons on each side. How would this be realized with biological neurons?

Summary: The authors show by approximate analysis of two identical continuous attractor networks, reciprocally coupled by Gaussian weights, that such a network can approximately implement the Bayesian posterior solution for queue integration.

Submitted by Assigned_Reviewer_5

"Reciprocally Coupled Local Estimators Implement Bayesian Information Integration Distributively" puts forward a new take on Bayesian integration of multimodal cues. Instead of assuming a special area in the brain, where evidence from various sensory cues is combined (as in Ma and all, 2006), the authors consider a scenario, whereby each area receiving direct afferent input from a single modality (i.e. visual) combines information from other modalities (e.g. vestibular) via reciprocal connections.
In the example analysed by the authors, and under a number of suitable assumptions, the cue integration they observe in their networks is close to Bayes-optimal.

Building up on work of Fung and all (2010), the authors derive theoretical predictions for the integration of information in reciprocally coupled ring attractors (CANNs), which they also confirm by simulations. The reader is led through the general steps of the analysis, while details are provided in the supplementary material.
The main result is that for a sufficiently weak input, and sufficiently narrow interactions (HWHH = 67 degrees for their example of heading direction), each of the networks combines information from the two cues in a Bayes optimal manner. That is, the decoding precision of a single network when both cues are available is a sum of this network's precisions when only a single cue is present, i.e.
p(theta| c1, c2) = p1(theta|c1) + p1(theta|c2),
where c1 is either
- the directly encoded cue, e.g. visual, which has a higher decoding accuracy, or
- the other modality cue, reported via reciprocal connections, with decoding accuracy growing with the strength of reciprocal connections.

This result is interesting despite the simplifying assumptions. My only major concern would be the lack of attention given to the posterior mean. As indicated by Fig. 3C, if the cues were not congruent, the networks would display bias towards the mean of the two cues larger than predicted by Bayesian inference. Although presented, this effect is not discussed anywhere in the article. Likewise, the Bayes estimate of the mean could be plotted together with the results of the simulations in Fig. 4.

Quality
This article is technically sound, supported well by theoretical predictions and simulations. It would be possible to evaluate weaknesses of the work in more detail (see above).

Clarity
The article is clearly written, although it would benefit from one more careful reading.
In Fig. 4 legend, values of mu_1 and mu_2 seem to be swapped. L. 395, reference to Eq. 3 was possibly meant to be Eq. 2.

Originality
To the best of my knowledge, the approach is new. The work it builds upon is well referenced.

Significance
As above, I believe the impact of this work will reach a wide audience, far wider than modellers working with attractor networks.
Summary: An interesting idea supported by rigorous analysis, and simulations.

Submitted by Assigned_Reviewer_7

SUMMARY

In this paper, the authors show that a continuous attractor network can perform cue combination in a Bayesian optimal way. More specifically, they consider a 2-ring model where each ring receives a noisy cue. They show analytically that when the two cues are present, the decoding reliability (which corresponds to the inverse of the stationary width of the Gaussian bumps on each ring) is given by the sum of the reliability of each cue, as one would expect from a Bayesian perspective.

The paper is well written and the main message is very elegant.

MAJOR

The authors use a representation of uncertainty which is consistent with a probabilistic population code (PPC). Consequently, the proposed approach will inherit all the advantages as well as the shortcomings of PPC (e.g. how would the framework scale for higher dimensional signals?) vs another way of representing uncertainty such as the sampling hypothesis (Fiser et al. TICS 2010). The authors might want to comment on this.


MINOR

- p3 after Eq. 6. Shouldn't $a_{ll} < \pi$ be replaced by $a_{lm} < \pi$
- "un-coupled" -> "uncoupled"
- "in the below" -> "in the text below"
Summary: The paper is well written and the main message is very elegant.
Author Feedback

Author rebuttal: Reply to Assigned_Reviewer_1:

We believe that the reviewer have missed the key contribution of our work. Here, our main contribution is to propose a new framework for implementing Bayesian information integration in the neural system. The previous models (e.g., Ma et al 2006) assume a centralized estimator (a special area in the brain) integrates information from multiple sensory cues. In this study, motivated by the experimental finding that both VIP and MSTd integrate the visual and vestibular cues for heading direction in a Bayesian optimal way despite they only receive a single cue directly (Gu, et al., 2008; Chen, et al., 2013), we propose a model demonstrating that Bayesian information integration in the brain can be achieved in distributively through reciprocal interaction between cortical regions. We believe that our contribution is novel and has potential important impact to the field. This was also acknowledged by other reviewers.

In response to the reviewer’s comments on the biological plausibility and technical issues of our model, our replies are presented below:

1. The reviewer’s claim that “the connection of our model to neuroscience is rather weak” is very unfair. In the paper, we have clearly described the biological relevance of each assumption used in the model. See lines 74-75 and 86-88. We did apply some mathematical tricks to solve Eqs.(2,3), but they are purely for the convenience of theoretical analysis and help us to get insight into understanding the network dynamics. They do not affect our conclusions, as confirmed by simulation.

2. The use of weak input limit is only for the convenience of theoretical analysis. Our result is actually applicable to a broad range of input strengths, as confirmed by simulation. See the caption of Fig.3, line 360, the input strength takes a wide range of values, \alpha=(0.1,1.5)U^0. These input amplitudes are at the sublinear region of the s-shaped f-I curve and they are strong enough to activate the network rapidly.

3. In the Fig.3B, we set μ_1=μ_2 to illustrate that decoding variance decreases for combined inputs. In Fig. 3C & D, the results for conflict and congruent cues are presented.

4. At this stage, limited by neurophysiologic data, it is difficult to quantitatively compare our model with experiment finding. Nevertheless, NIPS welcomes theoretical study inspired by experimental findings.

5. We have described the biological background of model assumptions in the paper. As for the fine tuning of CANN structure, it is a long-standing debate in the field. A few promising mechanisms have been proposed to overcome this difficulty, for instances, by including short-term facilitation in the recurrent connections (Itskov, Hansel and Tsodyks, 2011) and by reducing the inhibition strength around the bump area (Carter and Wang, 2007). A recent study has confirmed the anti-symmetrical shape of the correlation strength between MT neurons, a correlation structure uniquely associated with the CANN dynamics (Ponce-Alverez, et al.,PNAS 2013). Considering the tuning similarity between MSTd and MT neurons, it is reasonable to assume that the CANN is a suitable model for describing the neural circuit in MSTd.

6. Our model does not require more inter-network connections than the inner-network ones. In our study, the strength of inter-network connection J_{rp} is always weaker than the inner-network strength J_{rc} (see the caption of Fig.3, J_{rp}=(0.3,1)J_{rc} ). Furthermore, the abundant existence of reciprocal connection between MSTd and VIP has been well supported by the experimental data, including both anatomical (Boussaoud et al., 1990; Baizer, et al., 1991) and fMRI (Vincent, et al., 2007) evidences. Here, our study proposes a functional role of these reciprocal connections.

Reply to Assigned_Reviewer_5:

Thanks a lot for the encouraging comments of the reviewer.

Yes, the posterior mean is biased toward the mean of two cues, and the bias magnitude depends on the parameters. We will take the advice of the reviewer and explore the posterior mean of Bayesian integration in more detail in the revised paper.

Reply to Assigned_Reviewer_7:

Thanks a lot for the encouraging comments of the reviewer.

We will correct those typos in the revised manuscript.

Yes, it will be very interesting to explore the effect of reciprocal interaction on information integration in the sampling-based representation framework. Indeed, in our study after the network dynamics is simplified, it essentially becomes a set of coupled linear equations, Eqs.(14-15), and our result shows that positive reciprocal interactions have the effect of decreasing the variances of individual variables. Thus, in the framework of sampling-based representation, reciprocal interaction should have the similar effect on decreasing the variances of neuronal responses in the coupled neural ensembles.